# Hsa_circ_0015278 Regulates FLT3-ITD AML Progression via Ferroptosis-Related Genes

**DOI:** 10.3390/cancers15010071

**Published:** 2022-12-22

**Authors:** Jiquan Jiang, Jing Feng, Xiangnan Song, Qing Yang, Hongbo Zhao, Rui Zhao, Xinrui He, Yaoyao Tian, Lianjie Wang, Yanhong Liu

**Affiliations:** 1Department of Laboratory Diagnosis, The Second Affiliated Hospital of Harbin Medical University, Harbin 150086, China; 2Department of Hematology, The Second Affiliated Hospital of Harbin Medical University, Harbin 150086, China

**Keywords:** acute myeloid leukemia (AML), circRNAs, ferroptosis, ceRNA, FLT3-ITD

## Abstract

**Simple Summary:**

Acute myeloid leukemia (AML), especially the *FLT3-ITD* mutation subtype, has a poorer prognosis and higher risk of recurrence, which seriously threatens human health. Ferroptosis, an iron-dependent regulated cell death, is involved in the development and progression of AML. The mechanism by which circRNAs regulate the pathogenesis and prognosis of *FLT3-ITD* mutant-type AML through ferroptosis-related genes (FerRGs) remains unclear. In the present study, we aimed to decipher the pathogenic role of circRNAs in AML with *FLT3-ITD* mutation. Encouragingly, we discovered a circRNA, hsa_circ_0015278, that regulates ferroptosis-related genes by sponging miRNAs to promote *FLT3-ITD* AML progression. Furthermore, ferroptosis-related genes with prognostic value contribute to *FLT3-ITD* AML progression by regulating the tumor microenvironment. In conclusion, we constructed a ceRNA regulatory network involving hsa_circ_0015278/miRNAs/FerRGs, and the hsa_circ_0015278 signaling axis contributes to the identification of potential diagnostic and prognostic biomarkers and provides new insight into the pathogenesis and therapeutic targets of AML with *FLT3-ITD* mutation.

**Abstract:**

AML with the *FLT3-ITD* mutation seriously threatens human health. The mechanism by which circRNAs regulate the pathogenesis of *FLT3-ITD* mutant-type AML through ferroptosis-related genes (FerRGs) remains unclear. Differentially expressed circRNAs and mRNAs were identified from multiple integrated data sources. The target miRNAs and mRNAs of the circRNAs were predicted using various databases. The PPI network, ceRNA regulatory network, GO, and KEGG enrichment analyses were performed. The “survival” and the “pROC” R packages were used for K-M and ROC analysis, respectively. GSEA, immune infiltration analysis, and clinical subgroup analysis were performed. Finally, circRNAs were validated by Sanger sequencing and qRT-PCR. In our study, 77 DECircs-1 and 690 DECircs-2 were identified. Subsequently, 11 co-up-regulated DECircs were obtained by intersecting DECircs-1 and DECircs-2. The target miRNAs of the circRNAs were screened by CircInteractome, circbank, and circAtlas. Utilizing TargetScan, ENCORI, and miRWalk, the target mRNAs of the miRNAs were uncovered. Ultimately, 73 FerRGs were obtained, and the ceRNA regulatory network was constructed. Furthermore, MAPK3 and CD44 were significantly associated with prognosis. qRT-PCR results confirmed that has_circ_0015278 was significantly overexpressed in *FLT3-ITD* mutant-type AML. In summary, we constructed the hsa_circ_0015278/miRNAs/FerRGs signaling axis, which provides new insight into the pathogenesis and therapeutic targets of AML with *FLT3-ITD* mutation.

## 1. Introduction

Acute myeloid leukemia (AML) is a hematological malignancy caused by the arrest of differentiation, uncontrolled proliferation, and delayed apoptosis of immature myeloid progenitor cells and myeloid blast cells, which is highly clinically heterogeneous and aggressive [1]. It is a considerably common type of leukemia in adults [2]. According to the American Cancer Society, 20,050 new cases of AML are anticipated to be diagnosed in the United States by 2022, and 11,540 people will die from AML [3]. In addition, morbidity and mortality are on the rise with increasing age. Therefore, AML is a class of hematological malignancy that seriously threatens human health.

Fms Related Receptor Tyrosine Kinase 3, also known as FLT3, is often aberrantly expressed in hematopoietic tumors [4]. More than one-quarter of patients with AML have an internal tandem repeat mutation in the juxtamembrane domain of the FLT3 receptor (i.e., *FLT3-ITD* mutation), which promotes AML progression [5,6]. Compared with *FLT3-ITD* wild-type AML, *FLT3-ITD* mutant-type AML presented a higher risk of relapse and a significantly worse prognosis [7]. Hence, elucidating the underlying molecular mechanisms of the initiation and development of *FLT3-ITD* mutant-type AML will have important clinical significance for improving the diagnosis and treatment of AML with *FLT3-ITD* mutation.

Ferroptosis is a novel form of regulated iron-dependent cell death characterized by the accumulation of intracellular iron and lipid peroxidation [8]. In recent years, increasing studies have shown that ferroptosis participates in the occurrence and progression of various diseases, including cancer [9]. For example, lncRNA-PMAN was upregulated by HIF-1α and enhanced the stability of *SLC7A11* mRNA, which suppressed the accumulation of reactive oxygen species (ROS) and irons in gastric cancer cells, and thus protected gastric cancer cells from ferroptosis induced by Erastin and RSL3 [10]. The RNA-binding protein NKAP (NF-κB activating protein) protects glioblastoma cells from ferroptosis by binding to m^6^A and then promoting *SLC7A11* mRNA splicing and maturation [11]. Moreover, the transcription factor nuclear factor (erythroid-derived 2)-like 2 (NRF2), which plays a crucial role in cancer progression, was deubiquitinated by USP11, and the depletion of USP11 contributes to the suppression of cell proliferation and the induction of ferroptosis [12]. Nevertheless, the current studies on ferroptosis mainly focus on solid tumors, and its underlying pathogenic mechanism in AML is yet largely unknown and needs further elucidation.

Circular RNAs, as members of the noncoding RNA family, have attracted intense attention in recent years [13]. CircRNA is a covalently closed circular structure formed by back-splicing of mRNA precursors, which means that it is not easily degraded by exonuclease, is more stable than linear RNA, and has a much longer half-life than linear transcripts [14,15]. In addition, circRNAs also exhibit high sequence conservation and specificity for tissue development [16,17]. These characteristics make circRNAs ideal biomarkers and potential therapeutic targets for diseases such as cancer [16,18,19]. Accumulating evidence suggests that circRNAs are involved in the initiation and progression of various diseases by regulating a variety of biological processes, including gene expression and transcription [20,21]. Generally, circRNAs have been shown to act as a “sponge” for microRNAs, binding miRNAs through miRNA response elements (MREs), thus indirectly regulating the expression of miRNA target genes, which is known as the competitive endogenous RNA mechanism (ceRNA) [22,23,24]. For example, the highly expressed circRNA-DLEU2 acts as a “sponge” to adsorb miRNA-496 and enhance the expression of PRKACB, thereby promoting the proliferation and inhibiting the apoptosis of AML cells [25]. CircRNA_100290 competitively binds to miRNA-203, thereby up-regulating the expression of Rab10, a member of the RAS signaling pathway, to accelerate the progression of AML [26]. In addition, with the development of high-throughput sequencing and microarray technology, bioinformatics has been widely applied in tumor research. The understanding of circRNA expression profiles in various cancers has also been rapidly expanded [27]. However, few studies have been reported on circRNAs in AML with *FLT3-ITD* mutation subtypes, which have a poor prognosis.

In the present study, we aimed to decipher the pathogenic role of circRNAs in AML with *FLT3-ITD* mutation. Firstly, the circRNAs expression profiles of three cases of *FLT3-ITD* mutant-type AML, three cases of *FLT3-ITD* wild-type AML, and four cases of healthy controls were preliminarily evaluated by a circRNA microarray, and the differentially expressed circRNAs (DECircs) were screened. Subsequently, the targeted miRNAs and mRNAs were predicted using multiple databases. The ceRNA regulatory network involving ferroptosis-related genes (FerRGs) was obtained by taking the intersection with the predicted target mRNA, the Ferr-Genes, and transcriptome microarray data. In addition, the expression levels were verified by RT-qPCR in AML cell lines, and the hsa_circ_0015278/miRNA/FerRGs signaling axis was finally identified, providing a new therapeutic target and comprehensive perspective for clinical diagnosis and treatment.

## 2. Materials and Methods

### 2.1. Data Acquisition and Processing

TCGA- LAML (https://portal.gdc.cancer.gov/ (accessed on 21 January 2021) and corresponding normal tissue data in GTEx (https://gtexportal.org/home/datasets (accessed on 30 January 2021) were retrieved. After excluding inaccurate or missing clinicopathological data, 173 AML and 73 normal samples were included. RNAseq data with the TPM (transcripts per million reads) format was processed by the Toil process (Vivian J et al., 2017). Gene Expression Omnibus (GEO) [28] (http://www.ncbi.nlm.nih.gov/geo (accessed on 15 February 2021) is an international, public, functional genomic database. CircRNA and gene expression profile microarrays were obtained from the GEO database (Appendix A). CircRNA microarray GSE94591 (platform GPL19978, Agilent-069978 Arraystar Human CircRNA microarray V1) contained 3 *FLT3-ITD* mutation AML samples, 3 *FLT3-ITD* wild AML samples, and 4 healthy controls (HCs) [29]. The gene expression profiles microarray GSE65409 (platform GPL6947, Illumina HumanHT-12 V3.0 expression beadchip) contained 30 AML samples and 8 healthy controls (HCs) [30]. Based on the platform annotation information, probes were converted into the corresponding gene symbols. The ferroptosis genes (Ferr-Genes), which include drivers, suppressors, and markers, were downloaded from the FerrDb V2 website [31] (http://www.zhounan.org/ferrdb/current/ (accessed on 11 March 2021).

### 2.2. Identification of Differentially Expressed circRNAs (DECrics) and Differentially Expressed Genes (DEGs)

We removed or averaged the probe sets without the corresponding gene symbols or the genes with multiple probe sets, respectively. Then, the “limma” package [32] was applied to identify the differentially expressed circRNAs and mRNAs with a rigorous threshold of |log2 fold-change (FC)| > 1.0 and a false discovery rate (FDR) < 0.05 by comparing AML samples and HCs. Afterward, the Volcano plots and heatmaps of DECrics and DEGs were visualized.

### 2.3. Establishment of the ceRNA Regulatory Network in AML

The target miRNAs of the DECircs were predicted using the website tools of the Circular RNA Interactome (CircInteractome) [33] (https://circinteractome.nia.nih.gov/ (accessed on 14 April 2021), circbank [34] (http://www.circbank.cn/index.html (accessed on 22 April 2021) and circAtlas 2.0 [35] (http://circatlas.biols.ac.cn/ (accessed on 24 April 2021). The prediction of the target mRNAs of the miRNAs was performed using the website tools of TargetScan [36] (https://www.targetscan.org/vert_80/ (accessed on 12 May 2021), ENCORI [37] (https://starbase.sysu.edu.cn/ (accessed on 14 May 2021) and miRWalk [38] (http://mirwalk.umm.uni-heidelberg.de/ (accessed on 27 May 2021). Notably, in order to enhance the reliability of prediction, only target mRNAs and miRNAs that were consistently identified and predicted by more than two databases were selected. Ferroptosis-Related DEGs were defined as FerRGs by integrating and intersecting DEGs, Ferr-Genes, and predicted mRNAs. Venn [39] (http://bioinformatics.psb.ugent.be/webtools/Venn/ (accessed on 29 May 2021) was used to overlap more than two groups of molecules. The ceRNA network was constructed by FerRGs, potential target miRNAs to the DECircs, and corresponding DECircs. Cytoscape [40] (version 3.9.0) is an open-source bioinformatics software for visualizing and analyzing molecular interaction. Visualization of the ceRNA network was produced by Cytoscape 3.9.0 software.

### 2.4. Functional Enrichment Analysis

Gene Ontology (GO) and Kyoto Encyclopedia of Genes and Genomes (KEGG) pathway enrichment analyses of FerRGs were performed using the “clusterProfiler” [41] package and the “org.Hs.eg.db” package for ID conversion. GO terms were screened with adj. *p* value < 0.05, including three aspects: The cellular component (CC), molecular function (MF), and biological process (BP). The KEGG pathway terms with adj. *p* value < 0.05 were significantly enriched, elucidating utilities of the biological system and high-level functions. Furthermore, GSEA [42] was performed on MAPK3 and CD44 expression between the high and low subgroups to relate functional differences. For each analysis, we performed 1000 repetitions of gene set permutations. The enriched gene sets were chosen based on a false discovery rate (FDR) < 0.25 and adj. *p* < 0.05.

### 2.5. PPI Network Construction and Hub Gene Screening

The PPI network was built utilizing the Search Tool for the Retrieval of Interacting Genes (STRING [43] version 11.5, https://cn.string-db.org/ (accessed on 13 June 2021), which was an online tool that supplied a visualization of PPI networks. Networks were conducted based on gene expression, and the minimum required interaction score of 0.7 was considered statistically significant. The hub genes were filtered from the PPI using the “CytoHubba” [44] plug-in of Cytoscape 3.9.0 software with the 12 classes of ranking methods.

### 2.6. Survival and Prognosis Analysis

To estimate the value of the identified ceRNA network and determine the miRNAs and mRNAs related to prognosis, we downloaded the miRNA and mRNA expression profiles and clinical information of AML patients from the TCGA database. Kaplan–Meier (K-M) survival analyses of the hub genes and miRNAs in the ceRNA network were performed using the “survival” R package to determine the relationship with the overall survival (OS) of AML patients in the TCGA database and “survminer” for visualization. The ROC curve was developed to assess the diagnostic performance of hub genes employing the “pROC” [45] package. The area under the curve (AUC) was calculated to assess the predictive ability of the biomarker.

### 2.7. Immune Infiltration Analysis

By applying the ssGSEA (single-sample Gene Set Enrichment Analysis) built-in in the “GSVA” package, we quantified the relative tumor infiltration levels of immune cell types. The correlation of MAPK3 and CD44 expression with the markers of 24 tumor-infiltrating immune cells (aDC [activated DC]; B cells; CD8 T cells; Cytotoxic cells; DC; Eosinophils; iDC [immature DC]; Macrophages; Mast cells; Neutrophils; NK CD56bright cells; NK CD56dim cells; NK cells; pDC [Plasmacytoid DC]; T cells; T helper cells; Tcm [T central memory]; Tem [T effector memory]; Tfh [T follicular helper]; Tgd [T gamma delta]; Th1 cells; Th17 cells; Th2 cells; Treg) were estimated in the AML samples. The number of permutations was set at 100 to improve the deconvolution algorithm’s accuracy, and *p* < 0.05 was regarded as statistically significant.

### 2.8. Validation of Circrnas Relative Expression Level

Human FLT3-ITD^+^ cells (MOLM-13) and FLT3-ITD^−^ cells (HL60) were purchased from MeisenCTCC and Procell, respectively. AML cells and normal cells (bone marrow stem cells [BMSC]) were cultured in RPMI 1640 (Sigma) and DMEM/F12 (gibco), respectively, with 10% fetal bovine serum (gibco). All cells were cultured in a humidified atmosphere at 37 °C containing 5% CO_2_. The circular structure of circRNAs was confirmed by Sanger sequencing and divergent primer PCR. Total RNA was extracted from AML and normal cell lines using the TRIzol reagent (Invitrogen), and complement DNA was generated using the ReverTra Ace qPCR RT Master Mix with the gDNA Remover kit (TOYOBO). qRT-PCR assays were performed using the SYBR Green Realtime PCR Master Mix (TOYOBO). The circRNA expression levels were normalized to GADPH and each sample was tested in triplicate. The relative expression was analyzed by the 2^−ΔΔCt^ method. The primer sequences of circRNAs and GAPDH were synthesized by BGI (Beijing, China). The primers used in this study are listed in Appendix A.

### 2.9. Statistical Analysis

All analyses were performed using R v. 4.0.3 (https://www.R-project.org (accessed on 15 June 2021) and visualized by the “ggplot2” package v. 3.3.3 (https://www.rdocumentation.org/packages/ggplot2/ (accessed on 15 June 2021)). Data were presented as means ± standard error of the mean (SEM) in at least three independent experiments and analyzed with GraphPad Prism 9.3.1 (San Diego, CA, USA). The Wilcoxon rank sum test and the Weltch t’ test were used to compare the expression differences of unpaired samples between the two groups. The Wilcoxon rank sum test was utilized to analyze the relations between the clinicopathological features and candidate FerRGs and miRNAs. Spearman correlation analysis was used to present the degree of immune cell infiltration. *p* < 0.05 was considered statistically significant.

The detailed flowchart is exhibited in Appendix A. Bioinformatics tools and databases used in this study are listed in Appendix A.

## 3. Results

### 3.1. Identification of DECircs and DEGs in GEO Database

The circRNA microarray datasets (GSE94591) contain 10 samples, namely, 3 *FLT3-ITD* mutant-type (FLT3^+^), 3 *FLT3-ITD* wild-type (FLT3^−^) AML samples, and 4 healthy controls (HCs). DECircs-1 between *FLT3-ITD* mutant AML samples and *FLT3-ITD* wild AML samples were screened with the cutoff threshold |LogFC| > 1 and adj. *p* value < 0.05. Consistently, DECircs-2 between *FLT3-ITD* mutant AML samples and HCs were screened with the same criteria. As shown in the Volcano plot and Heatmap, 77 DECircs-1 containing 60 up-regulated and 17 down-regulated DECircs (Figure 1A,C) and 690 DECircs-2 containing 283 up-regulated and 407 down-regulated DECircs were identified (Figure 1B,D), respectively. Subsequently, the top 20 up-regulated DECircs-1 and DECircs-2 were selected and intersected (Figure 1E). Ultimately, 11 co-up-regulated DECircs were obtained for subsequent studies (11 DECircs were listed in Appendix A), and their structural patterns were shown in Figure 1F. Furthermore, the subcellular localization of circRNA was performed through the lncLocator online service website [46] (http://www.csbio.sjtu.edu.cn/bioinf/lncLocator/index.html (accessed on 5 July 2021). Surprisingly, except for hsa_circ_0060828 and hsa_circ_0049356, the remaining circRNAs were principally localized in the cytoplasm (Figure 1G). In addition, the same screening criteria were applied to analyze mRNA from the GSE65409 dataset. A total of 2832 DEGs contained 1237 up-regulated and 1595 down-regulated DEGs in the GSE65409 transcriptome expression profile (Appendix A).

### 3.2. Construction of the circRNAs/ miRNAs/FerRGs Regulatory Network

Many studies have reported that circRNA with abundant MREs (miRNA response elements) can regulate the expression of target genes by acting as a “sponge” for the adsorption of miRNA, thereby affecting the progression of the disease. For the sake of further elucidating the mechanism by which circRNAs affect the occurrence and progression of AML, we sought to establish a ceRNA regulatory network involving certain circRNAs. First, the bioinformatics software CircInteractome, CircBank, and circAtlas 2.0 were utilized to predict the target miRNAs of circRNAs. Then, the obtained miRNAs were used to predict the downstream target genes through TargetScan, ENCORI, and the miRWalk database. It is worth noting that only the miRNAs or target genes predicted by more than two prediction databases were considered credible and used for subsequent research. Simultaneously, 73 FerRGs were obtained by taking the intersection of DEGs, miRNA-predicted target genes, and ferroptosis genes downloaded from FerrDb (Appendix A). Ultimately, a ceRNA regulatory network involving circRNAs, miRNAs, and FerRGs with 599 edges and 109 nodes identified was visualized by utilizing Cytoscape 3.9.0 software (Figure 2A). Furthermore, to further elucidate the interaction between 73 FerRGs, the PPI network was analyzed by STRING and Cytoscape, respectively (Appendix A).

### 3.3. GO and KEGG Functional Enrichment Analysis of FerRGs

To better understand the mechanism of 73 FerRGs in AML, GO, and KEGG enrichment analyses were performed (the detailed enrichment information was listed in Appendix A). In terms of GO-BP (Figure 2B), the top three enriched biological processes were “response to metal ion” (GO: 0010038), “cellular response to oxidative stress” (GO: 0034599), and “cellular response to extracellular stimulus” (GO: 0031668). According to GO-CC (Figure 2C), the top three enriched cell components were “cell leading edge” (GO: 0031252), “pigment granule” (GO: 0048770), and “melanosome” (GO: 0042470). From the perspective of GO-MF (Figure 2D), the top three enriched molecular functions were “protein serine/threonine kinase activity” (GO: 0004674), “ubiquitin protein ligase binding” (GO: 0031625), and “coenzyme binding” (GO: 0050662). The KEGG results revealed that FerRGs were related to tumor and ferroptosis-associated pathways (Figure 2E), especially “ferroptosis” (hsa04216), “FoxO signaling pathway” (hsa04068), “mTOR signaling pathway” (hsa04150), “p53 signaling pathway” (hsa04115), and “mineral absorption” (hsa04978).

### 3.4. Screening and Enrichment Analysis of Hub Genes

The “Cytohubba” plug-in in Cytoscape (version 3.9.0) was used to screen hub genes. As is well-known, there are 12 algorithms in the “Cytohubba” plug-in (MCC, DMNC, MNC, Degree, EPC, BottleNeck, EcCentricity, Closeness, Radiality, Betweenness, Stress, and ClusteringCoefficient), and each algorithm can obtain the top 10 hub genes. Subsequently, we sorted the hub genes obtained by the 12 algorithms according to their frequencies and finally determined 15 hub genes (15 hub genes are shown in Appendix A, and the currently known roles that the hub genes identified in this study play in regulating the progression of *FLT3-ITD* mutated AML are listed in Appendix A.). The upset graph displayed the frequency of 15 hub genes identified by 12 algorithms (Appendix A). Notably, SIRT1, MAPK3, NFE2L2, CDH1, STAT3, and SQSTM1 were identified by more than eight algorithms, demonstrating their significance. In addition, to better understand the potential molecular function of hub genes, GO and KEGG enrichment analyses were carried out for 15 given hub genes. As shown in the cnetplot (Appendix A), GO enrichment analysis indicated that the 15 hub genes mainly participated in the processes of “cellular response to oxidative stress” (GO: 0034599), “myeloid cell differentiation” (GO: 0030099), “myeloid cell homeostasis” (GO: 0002262), and “protein serine/ threonine kinase activity” (GO: 0004674). KEGG pathway enrichment analysis revealed that the 15 hub genes were significantly involved in the “FoxO signaling pathway” (hsa04068), “ErbB signaling pathway” (hsa04012), “HIF-1 signaling pathway” (hsa04066), “Ferroptosis” (hsa04216), and “Acute myeloid leukemia” (hsa05221). Meanwhile, the ceRNA regulatory network of 15 hub genes was constructed. Ultimately, the regulatory network, which involved 11 circRNAs, 34 miRNAs, and 15 hub genes, was identified (Figure 3). Obviously, hsa_circ_0015278 exhibits sophisticated regulatory characteristics in the Sankey diagram. For this reason, it will be the focus of subsequent research.

### 3.5. Validation of Expression and Prognosis of 15 Hub Genes

TCGA (The Cancer Genome Atlas), which involves comprehensive and multidimensional data, including multi-omics and clinical data, is an essential database for Cancer researchers. The TCGA database was utilized for differential expression analysis of 15 hub genes (Figure 4A–O). The results showed that SIRT1, MAPK3, NFE2L2, CDKN1A, CDKN2A, CDH1, ATG7, CD44, and GSK3B exhibited significantly high expression in AML samples. Nevertheless, MAPK8, STAT3, TFRC, SQSTM1, G6PD, and SRXN1 exhibited significantly low expression in AML samples. Meanwhile, the consistency of differential expression of 15 hub genes was verified between TCGA and transcriptional expression profiling microarray GSE65409. According to the principles of the ceRNA regulatory mechanism, due to hsa_circ_0015278 being significantly over-expressed in AML samples, the mRNA regulated by circRNA should be similarly over-expressed in AML samples. Hence, six hub genes (MAPK3, CDKN1A, CDKN2A, ATG7, CD44, and GSK3B), which are highly expressed in GSE65409 and TCGA, were selected for subsequent study. Subsequently, six candidate hub genes were evaluated for diagnostic and prognostic efficacy using the TCGA database. It is worth mentioning that six candidate hub genes showed excellent diagnostic efficacy, and the area under the curve (AUC) was above 0.6. MAPK3 (AUC: 0.640, Cl: 0.573–0.707), CDKN1A (AUC: 0.657, Cl: 0.588–0.726), GSK3B (AUC: 0.796, Cl: 0.741–0.851), ATG7 (AUC: 0.873, Cl: 0.829–0.918), and even the AUC of CDKN2A and CD44 were approximately 1 (Appendix A). Notably, K-M (Kaplan–Meier) survival analysis showed that the expression of MAPK3 (Log-rank *p* = 0.022) and CD44 (Log-rank *p* = 0.001) were significantly associated with OS (Appendix A).

### 3.6. Validation of Relative Expression Levels and Structures of Candidate circRNAs

Next, we designed divergent primers and used qRT-PCR to verify the expression levels of four randomly selected circRNAs. As shown in Figure 5A–D, four candidate circRNAs (hsa_circ_0015278, hsa_circ_0049365, hsa_circ_0003162, and hsa_circ_0011536) were significantly highly expressed in HL-60 and MOLM-13 cells compared with BMSCs. In particular, the four circRNAs were more highly expressed in the FLT3^+^ AML cell line (MOLM-13 cell) compared with the FLT3^−^ AML cell line (HL-60 cell). In addition, to confirm the circular structure of the candidate circRNAs, Sanger sequencing was performed. We then cross-checked the Sanger sequencing results with the base sequences of circRNAs provided by circBase, which confirmed the existence of back-splicing sites in the Sanger sequencing results (Figure 5E–H). These results indicated that four candidate circRNAs were present and significantly highly expressed in the FLT3^+^ cell line.

### 3.7. Construction of ceRNA Sub-Network and Correlation Analysis of Clinicopathological Parameters

As mentioned above, hsa_circ_0015278 exhibited sophisticated regulatory properties, and MAPK3 and CD44 revealed excellent diagnostic and prognostic values. We constructed and visualized a regulatory sub-network involving hsa_circ_0015278 with Cytoscape (Figure 5I). In other words, five ceRNA regulatory axes consisting of hsa_circ_0015278, five miRNAs (hsa-miR-29b-1-5p, hsa-miR-1200, hsa-miR-1278, hsa-miR-217, hsa-miR-128-1-5p), and two hub genes (CD44 and MAPK3) were recognized. Next, the TCGA database was used to perform KM prognostic analysis for the five miRNAs (Appendix A). Surprisingly, the values of hsa-miR-1200 and hsa-miR-1278 were zero in more than half of the TCGA samples, which could not be divided into high- and low-expression groups, so KM analysis could not be carried out. In addition, the correlations between miRNAs and mRNAs involved in the sub-network and clinicopathological parameters were analyzed (Appendix A). Clearly, hsa-miR-29b-1-5p was significantly associated with age, poor prognosis, FLT3 mutation, and NPM1 mutation. In comparison, hsa-miR-217 and hsa-miR-128-1-5p were significantly associated with NPM1 mutations. MAPK3 and CD44 were significantly associated with age and FLT3 mutation, respectively.

### 3.8. GSEA Enrichment Analysis of MAPK3 and CD44

To understand the functional pathway of MAPK3 and CD44, GSEA was performed. As shown in Figure 6A–L, MAPK3 was positively correlated with several pathways, such as “Kegg_Acute_Myeloid_Leukemia”, “Wp_Ferroptosis”, “Wp_Hematopoietic_Stem_Cell_Differentiation”, “Kegg_Erbb_Signaling_Pathway”, “Pid_Foxo_Pathway” and “Pid_Hif1_Tfpathway”, etc. GSEA enrichment analysis of CD44 showed that pathways containing “Hallmark_Heme_Metabolism”, “Hallmark_G2m_Checkpoint”, and “Hallmark_E2f_Targets” were significantly negatively correlated with CD44 expression (Appendix A).

### 3.9. Immune Infiltration Analysis

Our study evaluated the correlation between MAPK3, CD44, and tumor-infiltrating immune cells in the AML microenvironment by applying the “GSVA” package. The relationship between hub genes and immune markers was elucidated by the Spearman correlation analysis. As depicted in Figure 7A, the expression of MAPK3 was significantly related to the infiltration of immune cells (B cells, Cytotoxic cells, Macrophages, Neutrophils, NK CD56bright cells, NK CD56dim cells, NK cells, Th1 cells, Th17 cells, Treg and CD8 T cells, etc.) into the AML microenvironment. As depicted in Figure 7B, the expression level of CD44 was significantly associated with the infiltrating levels of three immune cells, including pDC, Th17 cells, and Th2 cells. These results indicated that the expression of the target genes was associated with immune infiltration, which may affect the progression and prognosis of AML.

## 4. Discussion

AML is a clinically heterogeneous hematological malignancy characterized by a blockage of the hematopoietic progenitor and blast cell differentiation, which seriously threatens human health [1]. Despite significant progress in the research on AML in recent years, the 5-year survival rate for AML patients with the *FLT3-ITD* mutation remains low at only 20% [7]. Targeted therapies for *FLT3-ITD* mutation, especially FLT3 kinase inhibitors, have been developed and are being used presently to treat FLT3-mutated AML patients. However, resistance is emerging, and relapse caused by secondary TKD mutations is seriously hindering the therapeutic effect [47]. Therefore, there is an urgent requirement for new strategies to target *FLT3-ITD* and to develop new diagnostic and prognostic biomarkers.

CircRNAs are a new class of single-stranded circular non-coding RNAs formed by back-splicing without a 5′ cap and a 3′ Poly-A tail [48]. CircRNAs are abundant and highly conserved in tissues and cells, making them ideal biomarkers. Several studies in the literature have reported that circRNAs can indirectly regulate the expression of target genes by acting as a “sponge” for miRNAs, thereby affecting the progression of diseases [19]. In addition, ferroptosis is a novel form of programmed cell death discovered in recent years. It has been reported that ferroptosis is closely associated with the progression of various diseases, including cancer [9]. However, ferroptosis has been poorly studied in AML. This study aimed to identify a novel signaling axis of circRNAs/miRNAs/FerRGs with diagnostic and predictive prognostic value and to find novel diagnostic and therapeutic targets for AML patients with *FLT3-ITD*-mutation.

In the current study, we constructed a ceRNA regulatory network involving FerRGs using bioinformatic analysis methods to identify diagnostic and prognostic biomarkers of AML. First, two AML-related datasets, GSE65409 and GSE94591, were downloaded from the GEO database. DEGs and DECircs in AML were analyzed and obtained, respectively. Notably, subcellular localization of the lncLocator database found that the vast majority of DECircs were localized to the cytoplasm. The CSCD database found that all DECircs had MREs. It is well known that for circRNAs localizing to the cytoplasm, possessing MREs is required to regulate the ceRNA mechanism. Then, the target miRNAs and downstream target genes of DECircs were predicted using various databases. Subsequently, FerRGs were obtained by intersecting the predicted target genes, DEGs, and Ferr-Genes downloaded from the ferroptosis database.

Next, we constructed a ceRNA regulatory network involving FerRGs and performed GO and KEGG on 73 FerRGs. GO functional annotation results showed that FerRGs were mainly enriched in “myeloid cell differentiation”, “myeloid cell homeostasis”, and “iron ion binding”. In addition, the KEGG results showed that FerRGs were associated with tumors and ferroptosis pathways, especially “ferroptosis”, “FoxO signaling pathway”, “mTOR signaling pathway”, and “p53 signaling pathway”. Long et al. demonstrated that FLT3 inhibition induces upregulation of histone deacetylase 8 (HDAC8) via FoxO3-mediated transactivation in FLT3-ITD^+^ AML cells. Upregulated HDAC8 inactivates p53, leading to leukemia maintenance and drug resistance [49]. Sykes et al. observed that FoxO was active in approximately 40% of AML patients. FoxO inhibition results in myeloid maturation and subsequent AML cell death [50]. These results indicate that FerRGs play a crucial role in AML with *FLT3-ITD* mutation.

Meanwhile, we screened two molecules with prognostic values, MAPK3 and CD44, using various algorithms of the “Cytohubba” plug-in and clinical prognostic analysis. MAPK3, alias ERK1, and Liu et al. found that the reduction of nuclear factor erythroid 2-related factor 2 (NRF2) elicited phosphorylation of ERK1/2 and then activated ferritinophagy, which led to the degradation of ferritin heavy chain 1 (FTH1). The proteasome releases free iron ions to trigger ferroptosis in macrophages [51]. Gu et al. revealed that overexpression of the cancer stem cell marker CD44 promotes the interaction between SLC7A11 and OTUB1. CD44 expression inhibits ferroptosis in cancer cells in an OTUB1-dependent manner [52]. In addition, MAPK3 acts as a driver and CD44 acts as a suppressor in the FerrDb. These results suggest an essential role for MAPK3 and CD44 in ferroptosis. The results of GSEA enrichment analysis showed that MAPK3 was significantly positively correlated with several pathways, including “ferroptosis” and “acute myeloid leukemia”. It indicated that the target genes (MAPK3 and CD44) predicted by DECircs screened by FLT3 mutation or not may be involved in the initiation and progression of AML through ferroptosis.

Due to the upstream hsa_circ_0015278 of MAPK3 and CD44 exhibiting regulatory sophistication in the ceRNA network, we focused on hsa_circ_0015278 and constructed a sub-ceRNA regulatory network involving hsa_circ_0015278 in this study. The correlation analysis of clinicopathological parameters showed that the miRNAs and mRNAs in the sub-ceRNA regulatory network were closely related to clinicopathological parameters. In addition, accumulating evidence in the literature reveals that tumor-infiltrating immune cells affect the efficacy of chemotherapy and immunotherapy, thereby affecting patient outcomes [53,54]. It is generally believed that the immunosuppressive microenvironment of leukemia and the immune-evasive nature of AML blasts contribute to leukemogenesis and progression in AML [55]. Nahas et al. revealed that a targeted reduction of PD-1-expressing T cells results in enhanced leukemia-specific immunity [56]. Yan et al. elucidated that immune/stromal cell-associated genes significantly influence the prognosis of AML [57]. In our study, immune infiltration analysis using the “GSVA” package revealed that the target genes in the sub-ceRNA regulatory network were significantly correlated with the degree of infiltration of immune cells. We speculated that the sub-ceRNA regulatory network discovered in this study will contribute to improving the therapeutic efficacy of targeted therapy in AML *FLT3-ITD* mutation.

It is commonly believed that most of the circRNAs located in the cytoplasm can indirectly regulate mRNA expression through the ceRNA mechanism [58]. It has been reported that hsa_circ_0015278 regulates the progression of non-small cell lung cancer through the miR-1278/SOCS6 signaling axis [59]. Therefore, we speculate that hsa_circ_0015278 also affects the progression of the AML via a similar mechanism. qRT-PCR showed that 0015278 was highly expressed in AML with *FLT3-ITD* mutation, and most importantly, its downstream MAPK3 and CD44 played a vital role in both ferroptosis and AML. These suggest that hsa_circ_0015278 can be used as a novel biomarker with diagnostic and prognostic values in AML with *FLT3-ITD* mutation.

To the best of our knowledge, some researchers have used the circRNA microarray GSE94591 to conduct AML-related research. However, the vast majority focused on the differentially expressed circRNAs between the AML and healthy control groups. For the first time, differentially expressed circRNAs in *FLT3-ITD* mutations were focused on in our study. Furthermore, for the first time, we link the roles of circRNAs and ferroptosis-related genes in the progression of AML with *FLT3-ITD* mutation, providing a new perspective for targeted therapeutic strategies. However, we only detected the expression of hsa_circ_0015278 in AML cell lines, lacking validation in a large number of clinical samples. Moreover, in-depth mechanistic studies are lacking. Therefore, in the future, we will explore the underlying molecular mechanism of hsa_circ_0015278-mediated disease progression of AML with *FLT3-ITD* mutation through a series of functional experiments and animal models.

## 5. Conclusions

In summary, based on GEO and TCGA database mining, bioinformatics analysis, and in vitro validation, we constructed a potential ceRNA regulatory network involving hsa_circ_0015278/miRNAs/FerRGs in AML with *FLT3-ITD* mutation. Encouragingly, MAPK3 and CD44, as hub genes, were not only enriched in “ferroptosis” and “acute myeloid leukemia” pathways but also significantly correlated with prognosis and immune cell infiltration in AML patients. Therefore, the hsa_circ_0015278 signaling axis contributes to the identification of potential diagnostic and prognostic biomarkers and provides new insights into the pathogenesis and therapeutic targets of AML with *FLT3-ITD* mutation and poor prognosis.

## Figures and Tables

**Figure 1 cancers-15-00071-f001:**
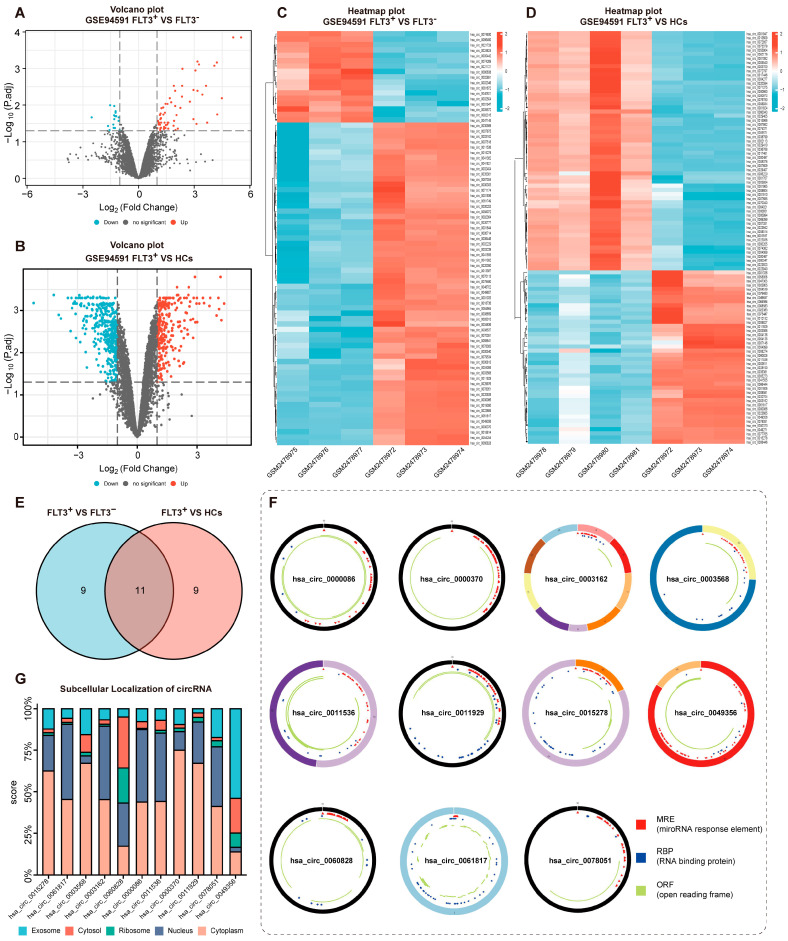
Identification of DECircs in AML. (**A**) Volcano plot of DECircs-1 from GSE94591 (FLT3^+^ VS FLT3^−^). (**B**) Volcano plot of DECircs-2 from GSE94591 (FLT3^+^ VS HCs). The red and blue dots represent up-regulated and down-regulated circRNAs, respectively. The vertical lines indicate two-fold upregulation and downregulation. The horizontal line represents adj. *p* = 0.05. (**C**) Heatmap of DECircs-1 from GSE94591 (FLT3^+^ VS FLT3^−^). (**D**) Heatmap of DECircs-2 from GSE94591 (FLT3^+^ VS HCs). Red and blue indicate high and low expression, respectively, and the color scales represent expression values. The columns represent samples, and the rows represent circRNAs. (**E**) Venn diagram of the intersection of the top 20 DECircs-1 and DECircs-2. (**F**) CSCD presents structural patterns of 11 circRNAs. (**G**) Subcellular localization of 11 DECircs. FLT3^+^, *FLT3-ITD* mutant type; FLT3^−^, *FLT3-ITD* wild type; HCs, healthy controls; DECircs, Differentially Expressed circRNAs.

**Figure 2 cancers-15-00071-f002:**
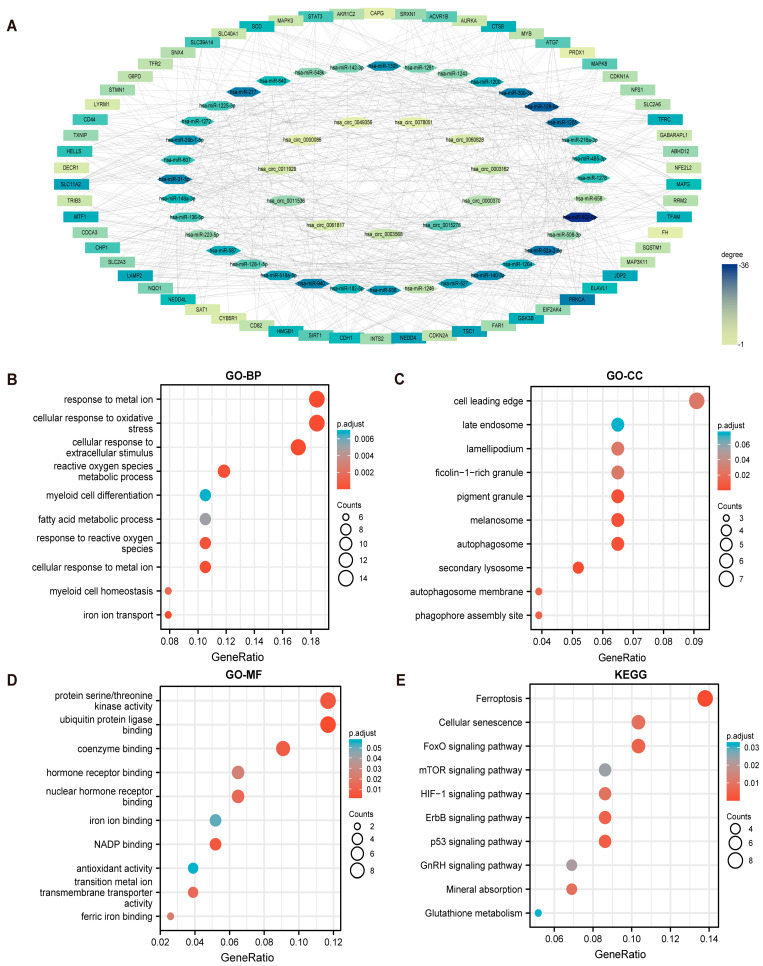
Construction of ceRNA regulatory network and GO and KEGG enrichment analysis. (**A**) The ceRNA regulatory network of circRNAs−miRNAs−-73 FerRGs in AML. Ovals indicate circRNAs, hexagons indicate miRNAs, and rectangles indicate FerRGs. The shade of color represents the degree of connectivity. (**B**–**E**) GO and KEGG functional enrichment analyses of 73 FerRGs in the ceRNA network. (**B**) The GO enrichment of BP. (**C**) The GO enrichment of CC. (**D**) The GO enrichment of MF. (**E**) The enrichment analysis of KEGG pathways. ceRNA, competitive endogenous RNA; BP, biological processes; CC, cellular components; MF, molecular functions.

**Figure 3 cancers-15-00071-f003:**
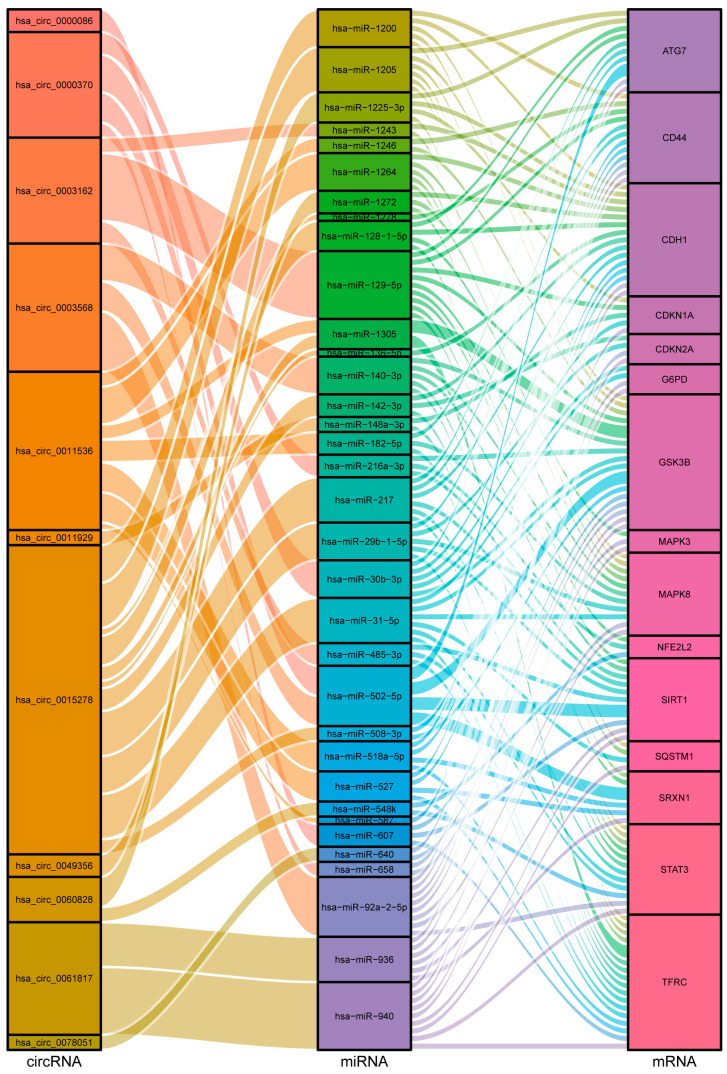
Sankey diagram presents the ceRNA network involving 15 hub genes. The area of the squares in the Sankey diagram represents the data content. Columns represent the molecular category, and different squares represent different molecules. The thickness of the lines represents the degree of closeness between the molecules.

**Figure 4 cancers-15-00071-f004:**
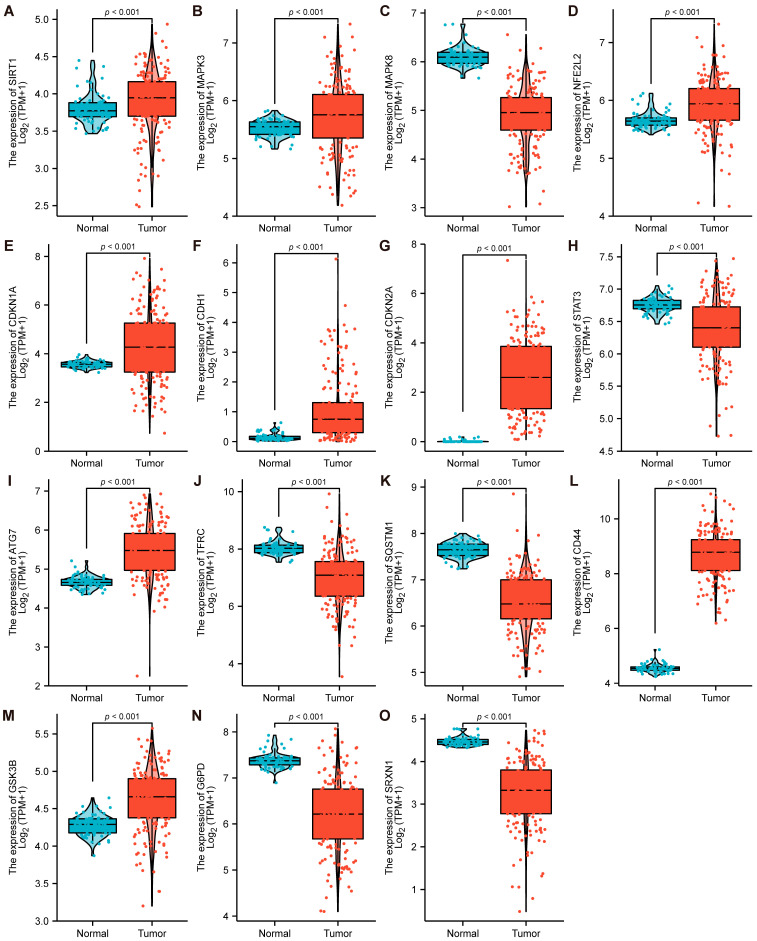
Validation of expression level of 15 Hub genes. (**A**–**O**) Differential expression analysis of 15 hub genes in AML and normal samples utilizing the TCGA database. The *p* value was calculated using a Wilcoxon rank sum test.

**Figure 5 cancers-15-00071-f005:**
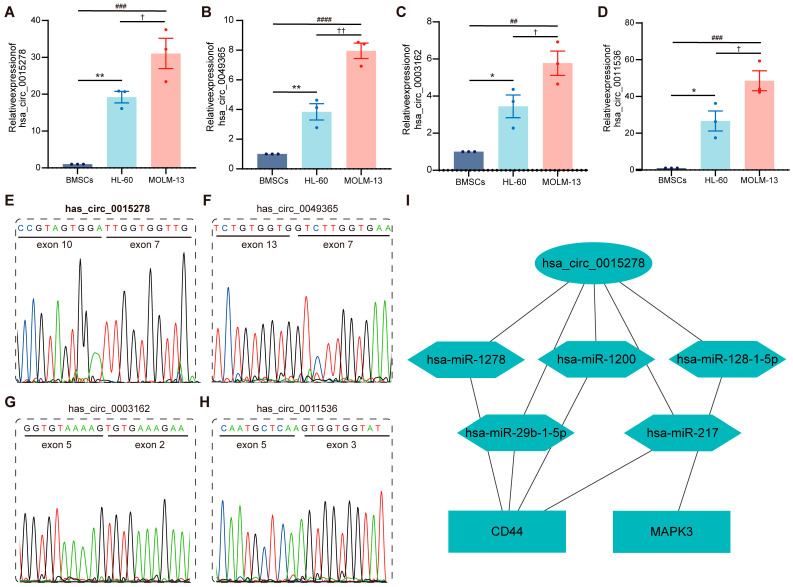
Validation of relative expression levels and structures of candidate circRNAs. (**A**–**D**) Validation of relative expression levels of circRNAs using qRT-PCR (**A**, hsa_circ_0015278; **B**, hsa_circ_0049365; **C**, hsa_circ_0003162; and **D**, hsa_circ_0011536). The *p* value was calculated using a Wilcoxon rank sum test. * *p* < 0.05 vs. BMSCs, ** *p* < 0.01 vs. BMSCs; ^##^
*p* < 0.01 vs BMSCs, ^###^
*p* < 0.001 vs BMSCs, ^####^
*p* < 0.0001 vs BMSCs; ^†^
*p* < 0.05 vs. HL-60, ^††^
*p* < 0.01 vs. HL-60. (**E**–**H**) Sanger sequencing confirmed back-splicing sites (E, hsa_circ_0015278; F, hsa_circ_0049365; G, hsa_circ_0003162; and H, hsa_circ_0011536). (**I**) Sub-network involving hsa_circ_0015278 was constructed and visualized with Cytoscape.

**Figure 6 cancers-15-00071-f006:**
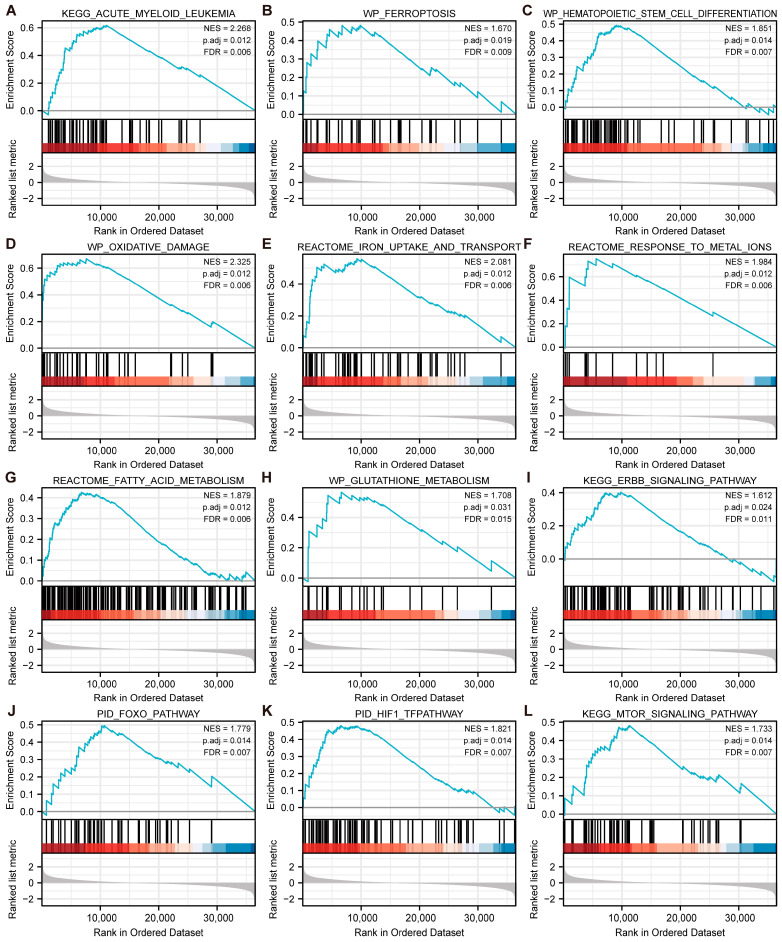
GSEA enrichment analysis of MAPK3. (**A**–**L**) Identification of the biological functions of MAPK3. Upregulated genes are located on the left, approaching the origin of the coordinates, whereas downregulated genes are on the right of the X-axis. Only gene sets with false discovery rate (FDR) < 0.25 and adj. *p* < 0.05 were considered significant.

**Figure 7 cancers-15-00071-f007:**
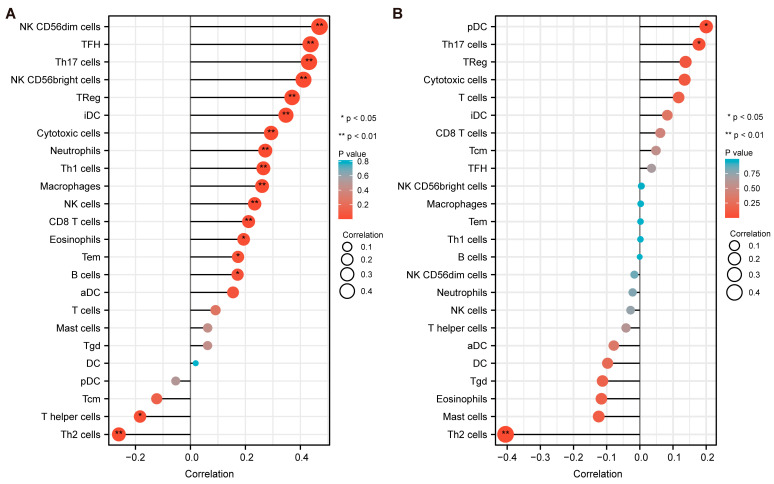
Immune infiltration analysis applying the “GSVA” package. (**A**) The relationship between MAPK3 and immune markers by Spearman correlation analysis. (**B**) The relationship between CD44 and immune markers by Spearman correlation analysis. The size of the dots indicates the absolute value of Spearman’s r. The larger the dot and the longer the bar, the higher the degree of correlation. * *p* < 0.05, ** *p* < 0.01.

## Data Availability

The data presented in this study are available in this article (and Appendix A).

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
