# Peer review of "Hsa_circ_0015278 Regulates FLT3-ITD AML Progression via Ferroptosis-Related Genes"

_cancers, 2022, doi:10.3390/cancers15010071_

Round 1

Reviewer 1 Report

The authors studied the pathogenic role of circRNAs in AML with FLT3-ITD mutation. Base on evaluatin circRNA microarray, and looking into screening the differentially expressed  circRNAs (DECircs). And then the targeted miRNAs and mRNAs  were predicted through multiple databases.

TCGA- LAML. Have normal tissues samples from the same patients, those could be separated and compared in addition to the comparison with GTEX tissues.

There is also a number of studies have data with FLT3-ITD mutations.  

and as another control, the authors could pick up data with other mutations (such as RUNX1 or NPM1 or others) and compare results if theses are same/different network outcome to the ones with FLT3-ITD mutation.

Author Response

December 13, 2022

Dear Reviewer:

Thank you very much for your time involved in reviewing the manuscript and your very encouraging comments on the merits.

We hope that the explanation and response have fully addressed all of your concerns.

In the attachment, we discuss each of your comments individually along with our corresponding responses.

Encl. Responses to the comments. Please see the attachment.

We would like to take this opportunity to thank you for all your time involved and for this excellent opportunity for us to improve the manuscript. We hope you will find this revised version satisfactory.

Kind regards,

Jiquan Jiang

Reviewer 2 Report

Please see my minor and major comments

1.     Line 26, grammar: Differentially expressed circRNAs and mRNAs were identified from multiple integrated data-sources, respectively.

2.     Line 410, grammar: Targeted therapies for FLT3-ITD mutation, especially FLT3 kinase inhibitors, have been developed and are being used presently to treat FLT3-mutated AML patients. However, resistance is emerging, and relapse caused by secondary TKD mutations is seriously hindering the therapeutic effect

3.     Line 417, grammar: Several studies in the literature have reported that circRNAs can indirectly regulate the expression of target genes

4.     Line 432, grammar: It is well known that for circRNAs localizing to the cytoplasm, possessing MREs is required to regulate the ceRNA mechanism

5.     Line 496, grammar: Accumulating evidence in literature reveals

6.     Line474, grammar: improve the therapeutic efficacy of targeted therapy in AML FLT3-ITD+ mutation

Major comments

1.     Line 27, addition: “target miRNAs and mRNAs were predicted”, how successfully were they predicted? Were there controls? What were those quality controls? Mention here

2.     Addition: Split Figure S1 into two parts. Too much information, making it difficult to follow. The first part should go up to the identification of 15 hub genes. Then, talk separately about enrichment analyses and the K-M prognostication

3.     Addition: The paper fluidly switches between various noncoding RNA molecules. Make a small flowchart/diagram at the end of the introduction that explains the various RNAs involved and how they are incorporated into the ultimate ceRNAs. Essentially pictorially demonstrate the following connection: circRNAs --> DECircs --> DEGs --> Ferroptosis Related DEGs (FerRGs) --> ceRNA network to project miRNAs to corresponding DECircs

4.     Addition: This project heavily relies on bioinformatics tools. Make a table at the beginning of the Methods section explaining each tool and its role in the ultimate analysis. For instance, TargetScan and miRWalk were used for the prediction of the target mRNAs of the miRNAs. Provide hyperlinks here in the table instead of throughout the text.

5.     Line 298: SIRT1, MAPK3, NFE2L2, CDH1, STAT3, and SQSTM1 were identified by more than 8 algorithms. Many of these hub-genes identified in the study have been reported in literature to participate in AML-progression. For instance, SIRT1 plays a major role in harboring activating mutations (https://ashpublications.org/blood/article/124/1/121/33146/SIRT1-prevents-genotoxic-stress-induced-p53). Comment on currently known role of these hub-genes in a table along with reported mutations and their influence on AML. This will make the transition to section 3.5 smoother as well

6.     Line 369: Comment on why they could have been zero, any potential explanations?

7.     Line 373: In recently reported literature, multivariate Cox regression identified FLT3-ITD, RUNX1, and TP53 mutations as independent prognostic factors. Any other associations of circRNAs aside from NPM1? (https://www.ncbi.nlm.nih.gov/pmc/articles/PMC9152364/)

8.  Line 400: Discuss the role that immune infiltration plays in AML, provide background, why is this significant for AML progression

Author Response

December 14, 2022

Dear Reviewer:

We are very grateful to Reviewer for reviewing the paper so carefully and hope that the explanation and response have fully addressed all of your concerns. As your suggestions, I seriously modified the grammatical spelling errors in the manuscript. Besides, I modified the format as required by the editor.

In the attachment, we discuss each of your comments individually along with our corresponding responses.

Encl. Responses to the comments. Please see the attachment.

We would like to take this opportunity to thank you for all your time involved and this excellent opportunity for us to improve the manuscript. We hope you will find this revised version satisfactory.

Kind regards,

Jiquan Jiang

Reviewer 3 Report

The Paper is very interesting, however, it is prepared in a form that is very difficult to understand. Authors should note that this journal publishes papers ranging from clinical data to advanced molecular biology combined with bioinformatics computations. In my opinion, the article is written in a way that does not prevent its understanding, even by a person dealing with experimental oncology or molecular biology. The article contains too many "shortcuts" and unexplained abbreviations. Descriptions under the figures are definitely too short and do not explain their message. Authors should prepare a list of abbreviations and their explanations, as the reader may be unable to remember them all while reading the paper. In addition, the descriptions of the procedures for obtaining data from databases are incomprehensible. The description of Figure 1c is pixelated and, therefore, illegible.

Author Response

December 14, 2022

Dear Reviewer:

Thank you very much for your time involved in reviewing the manuscript and your very encouraging comments on the merits. We are very grateful to the Reviewer for reviewing the paper so carefully and hope that the explanation and response have fully addressed all of your concerns.

According to your comments, we have further improved our manuscript. In particular, the result and conclusion have been refined. Your suggestion makes the manuscript improved and perfect. Besides, I modified the format as required by the editor.

In the attachment, we discuss each of your comments individually along with our corresponding responses.

Encl. Responses to the comments. Please see the attachment.

We would like to take this opportunity to thank you for all your time involved and this excellent opportunity for us to improve the manuscript. We hope you will find this revised version satisfactory.

Kind regards,

Jiquan Jiang

Round 2

Reviewer 2 Report

Your efforts have significantly improved the flow and clarity of the manuscript.

Your attention to detail and willingness to make necessary changes have not gone unnoticed.

Reviewer 3 Report

The changes introduced by Authors are satisfactory